# Approximating Optima of Nonconvex Functions

## Abstract

We study the computability of approximating optima of non-convex functions. We give a simple proof to show that the problem of finding the optimal value (and optimal point) or its approximation is not even computable in the oracle setting. We also give a property a function has to satisfy if its global optima can be approximated. Next we give an example of such a global property we call basin of attraction. Then we give a simple algorithm which converges to the global optima when this is known. Finally, we give some numerical results.

## 1 Introduction and Preliminaries

The problem of computing the global minima of a non-convex continuous function $f : C \to \mathbb{R}$, where $C \subset \mathbb{R}^d$ is a closed, compact subset is well studied (Horst & Tuy). . Global minima is the point $x^* \in C$ which satisfies the following property: $f(x^*) \le f(x)$ for all $x \in C$. The function $f$ attains this minimum at least once by extreme value theorem. Our goal is to find one such point.

We note that we consider an oracle setting, where the function values are given by an oracle. This is different from the computablity of optima computable real functions studied for example in (Pour-El & Richards, 1989). We would like to point to the fact that our oracle setting is more general than one considered for example in (Lee et al., 2023) and suitable for any non-convex optimization problem. In their setting, it is shown that the optimal value may be computable but the optimizer (optimal point) is not computable. We show that neither the optimizer nor the optimal value is computable in our setting.

The problem of non-convex minimization can be shown to be NP-hard by reducing the NP-complete subset-sum problem to a non-convex optimization problem (Murty & Kabadi, 1987). Let us define $S = \{x |, |f(x) - f(x^*)| \le \epsilon\}$ and term the members of $S$ as $\epsilon$-optima. In optimization literature (Foster et al., 2019; Zhang et al., 2020), it is also known that finding approximations to optima is not tractable for non-convex functions. For non-convex functions, $\epsilon$-stationary point which is weaker than $\epsilon$-optima is also known to be not tractable (Zhang et al., 2020). We show more in this paper, that this set $S$ is not computable. This is much stronger than saying they it is intractable.

It is easy to see that a simple grid search will output a sequence of points converging to the global optima. And for a Lipschitz continuous function, it requires an exponential number of oracle calls (Nesterov) if the Lipschitz constant is known. We show that Lipschitz constant is an example of a property a function must satisfy if its approximate optima is computable. We also see that if the Lipschitz constant is not known the (approximate) optima is not computable.

Our main contributions in this paper are:

1. An oracle setting which is applicable to many real-world optimization problems and different from the computable real-functions setting studied in computable analysis (Pour-El & Richards, 1989).

2. In the oracle setting we consider, the problem of approximating both the minimal value and the minimizer of non-convex functions or their approximations are not computable.

3. A simple property the function has to satisfy if its global optima can be approximated.

4. We give an example of global optima property-basin of attraction. And if this is known, we give an algorithm which converges to the global optima.

We now start with definition of the standard Turing machine here:

**Definition 1.1.** Turing machine has a three infinite tapes divided into cells, a reading head which scans one cell of the tape at a time, and a finite set of internal states $Q = \{q_0, q_1, \ldots, q_n\}$, $n \geq 1$. Each cell is either blank or has symbol 1 written on it. In a single step the machine may simultaneously (1) change from one state to another; (2) change the scanned symbol $s$ to another symbol $s' \in S = \{1, B\}$; (3) move the reading head one cell to the right (R) or left (L). This operation of machine is controlled by a partial map $\Gamma : Q \times S^3 \to Q \times (S \times \{R, L\})^3$.

*Remark* 1.2. The map $\Gamma$ viewed as a finite set of quintuples is called a Turing program. The interpretation is that if $(q, s_1, s_2, s_3, q', s_1', X_1, s_2', X_2, s_3', X_3) \in \Gamma$, in state $q$, scanning symbols $s_1, s_2, s_3$ changes state to $q'$ and in the tape $i$ input symbol to $s_i'$ and moves to scan one square to the right if $X_i = R$ (or left if $X_i = L$.) in the tape $i$.

## 1.1 Computable Numbers

**Definition 1.3.** A function $f : \mathbb{N} \to \mathbb{N}$ is called computable if there exists a Turing machine, given an input $x \in \mathbb{N}$ writes the value $f(x)$ in its output tape.

Rational numbers are assumed to be computable. We state the standard definition of a computable numbers, (Lee et al., 2023).

**Definition 1.4.** A sequence of rational numbers $(r_n)_{n \in \mathbb{N}}$ is said to be computable, if there exist recursive function $s, p, q : \mathbb{N} \to \mathbb{N}$ such that

$$r_k = (-1)^{s(k)} \frac{p(k)}{q(k)},$$

for all $k \in \mathbb{N}$.

**Definition 1.5.** A sequence of real numbers $(x_n)_{n \in \mathbb{N}}$ converges effectively to a limit $x \in \mathbb{R}$, if

$$|x_n - x| \leq 2^{-n},$$

for all $n \in \mathbb{N}$.

**Definition 1.6.** A real number $x \in \mathbb{R}$ is called computable, if there exists a rational computable sequence $(r_n)_{n \in \mathbb{N}}$ such that it converges effectively to $x$. The sequence $(r_n)_{n \in \mathbb{N}}$ is called a representation of $x$.

We can see that the set of computable numbers is closed under addition, subtraction, multiplication, division and effective convergence.

## 1.2 Finite Precision Representation

We assume for formality finite-precision numbers. This assumption of finite-precision numbers is useful to model real-world systems and is not restrictive. Consider any real number $x \in \mathbb{R}$. Let $r_0$ be the largest integer such that $r_0 \leq x$. Having chosen $r_0, r_1, \ldots, r_{k-1}$ choose largest positive integer $r_k$ such that

$$r_0 + \frac{r_1}{10} + \frac{r_2}{10^2} + \ldots \frac{r_k}{10^k} \leq x.$$

This is the decimal expansion of the number. We can check that this expansion is unique. We define precision length to be the number $k$. Now for finite precision representaion of a real we need to specify this precision length $k$. And we say for any real $x \in \mathbb{R}$, the numbers $r_0, r_1, \ldots, r_k$ is its finite precision representation. Note that $r_i, 0 \leq i \leq k$ can be zero. For a point $x \in \mathbb{R}^d$, given a precision length $k$ we can have decimal expansions for all its co-ordinates. Note that though we give binary representations to the Turing machine, for simplicity we assume precisions denote the decimal precisions. It can be shown that a number is computable if and only if there is a Turing machine which can give its finite precision representation for any precision $k$.

*Remark* 1.7. Suppose $r_1, \ldots, r_k$ is the finite precision representation with length $k$ of some real $x$. Let $\bar{x}$ be the number with decimal expansion $r_1, \ldots, r_k$ as $x$ and $r_l = 9$ for $l \geq k+1$. And let $\underline{x}$ be the number with decimal expansion $r_1, \ldots, r_k$ as $x$ and $r_l = 0$ for $l \geq k+1$. And the length of this interval $[\underline{x}, \bar{x}]$ is $\epsilon = 10^{-k}$. We then say with precision length $k$ we can represent consecutive numbers with gap greater than or equal to $\epsilon$.

## 1.3 THE PROBLEM

We assume there is an oracle for our continuous function $f$. This oracle gives the value $f(x)$ up to any finite-precision for a given finite-precision $x$. The Turing-machine has access to this function oracle. We also give a value $\epsilon > 0$ as input to the Turing machine. Our main problem is to write any point $x_o$ of the finite precision length such that $|f(x_o) - f(x^*)| < \epsilon$ i.e., it should find $\epsilon-$approximation of the global optima. We show that this problem is not computable.

Let us assume we have a three-tape Turing machine, one is used for calculations, second is for the giving the finite precision real and the precision length required to the function oracle and third one has the value returned from the oracle (Soare). Note that the third tape can also store the previous values. That is suppose we start with $x_0$ and find $x_1, \ldots, x_k$ this tape can store all these and also the corresponding function values obtained from the function oracle $f(x_0), \ldots, f(x_k)$ for finding $x_{k+1}$.

We consider these problems in this paper and answer them in negative.

**Problem 1.8.** *Given a continuous, non-convex function $f$, is there a Turing machine with access to the function oracle which can compute the global optimal value $f(x^*)$ of the function $f$ ?*

**Problem 1.9.** *Given a continuous, non-convex function $f$, is there a Turing machine with access to the function oracle which can compute the global optimal point (optimizer) $x^*$ of the function $f$ ?*

**Problem 1.10.** *Given a continuous, non-convex function $f$, is there a Turing machine with access to the function oracle which can compute a $\epsilon-$ approximation to the global optimal value and optimizer of the function $f$ ?*

## 1.4 REAL-WORLD APPLICATIONS

Global optimization has very wide applications ranging from science and engineering to finance. Let's look at the portfolio optimization problem in finance first. The standard objective here is to minimize a risk term measured by the variance of the rate of return of the portfolio subject to a constraint on the level of expected return. Though in this paper we have considered unconstrained global optimization, the problem will still be unsolvable if there were constraints.

Classical portfolio optimization can be formulated as a convex minimization problem and there are fast algorithms to solve it. When we extend it to include transaction costs, tax, and market impact which could be significant, we will get a non-convex problem. And we would have many local minima and the result proved here would apply. Let us now look at a couple of other real-world applications in some detail.

### 1.4.1 SUPERVISED LEARNING

In supervised learning algorithms including Neural Networks (NN) there is a loss or error function that needs to be minimized. In real-life problems like image classification, we have a rugged function, having many local optima. Hence simple methods like gradient search can not find the global optima. And many state-of-the-art NN architectures like CNN's still use the backpropagation algorithm which is based on gradient search and can not converge computably to the global minima as shown here. Hence we cannot give an algorithm to train the NN in the best possible way.

Now let us consider a supervised learning example of classifying handwritten digits (Bishop, 2006). We are given images $x$, $x \in D_T$ of the digits 0 to 9. The image $x$ is itself a matrix $M$ of individual pixel values. We have class labels from 0 to 9. We denote the class label of image $x$ by $C(x)$. We are given a set $D_T$ containing $N$ images and its class labels i.e., $D_T = \{(x_i, C(x_i)) \mid i = 1, \ldots N\}$. This set $D_T$ is called the training set. For example images, $x_1$ and $x_{10}$ may belong to a class labeled 5, $C(x_1) = C(x_{10}) = 5$.

The problem is to generalize and classify images which are not yet seen. That is we need to find the class of images $x$ that come from what is called a test set $D_E$ which are not present in the training set $D_T$. But, there are be some similarities between images belonging to the same class which we need to learn. A simple way to do this is by first fitting a function $y(x) = f(x, W)$ which gives the class labels of these images in the training set, i.e., $C(x) = y(x)$, $x \in D_T$. And then using probability by adding a noise term to the predicted output. Here the function $f(\cdot)$ is called the activation function

and is non-linear. Here $W$ is some parameter vector and models of the form $f(W^T x)$ are called linear models.

One can obtain closed-form solutions if certain assumptions like additive Gaussian noise are made. But in practice, the accuracy of these simple methods is much lower compared to NN's or Support Vector Machines. The reason is that the strong assumptions made do not hold in practice. If we relax these and use a simple gradient search we could easily get stuck in local optima without models like NN's. Thus the result of this paper justifies the need for these state-of-the-art models in practice.

### 1.4.2 CHEMICAL PROCESS OPTIMIZATION

Chemical process optimization is itself a research area with several problems (Edgar & Himmelblau). We give an example of one such problem. Assume that a chemical manufacturer needs to deliver to several customers located at different locations a single product. The manufacturer also has several plants. Then the problem here becomes determining the quantities to be manufactured at these plant locations so that the total cost is minimized.

Here we have $m$ plants and $n$ demand points and we need to find $Y_{ij}$, $i = 1, \ldots, m$, $j = 1, \ldots, n$. This quantity indicates how much we must produce at each plant $i$, $Y_i$ and how much of it should go to a particular demand point $j$, $Y_{ij}$. The cost-minimizing solution to this involves many factors transportation cost, production cost, and capacity curves. Some plants can also be more efficient than others. Hence we would have a unique functionality between production cost and production rate. We also need to count the transportation costs which would make the objective in the optimization problem complex/multi-modal (Edgar & Himmelblau). And we need to find the global optima. Even if one of the terms in this cost objective function is not known analytically because of uncertainty, as we prove there is no algorithm to compute even an approximation to it.

These are just a few broad examples and there are many such real-world applications to finding global optima where the function is known only by an oracle.

## 2 MAIN THEOREM

Given the objective function $f$, let the set of global minima be denoted by $G^f$. Now consider $\epsilon$-approximation to the global minima.

**Lemma 2.1.** *For all $\epsilon > 0$ there exists a point $x_n^*$ of finite precision length $n$ such that $|f(x^*) - f(x_n^*)| < \epsilon$.*

*Proof.* Let $\delta > 0$ be such that $|x - y| < \delta$ implies $|f(x) - f(y)| < \epsilon$. Such an $\delta > 0$ exists for all $\epsilon > 0$ because the function $f$ is continuous. Let $n$ be the precision length required to represent numbers with gap $\epsilon/10$ between consecutive numbers (Remark 1.7). Then we see for the global minima $x^*$ (like for all other points) its finite precision representation $x_n^*$ with precision length $n$ is such that $|x_n^* - x^*| < \epsilon$. $\qquad \square$

**Definition 2.2.** Let $G_{\epsilon,k}^f$ be the set of points with given finite precision length $k \geq 1$ where the function value is $\epsilon > 0$ close to the global minima. And $G_\epsilon^f$ be the union of all such sets.

We consider only finite-precision numbers. As there are only finite number of points with precision length $k$, the set $G_{\epsilon,k}^f$ is finite. Since we would like an algorithm to computably converge to a single point, for simplicity we assume the global optima is unique i.e., $G^f$ is a singleton. This is not uncommon in optimization literature as strict convexity gives unique local (global) minima and is assumed for objective functions. Now we need few results before we state the main theorem.

**Lemma 2.3.** *The problem of checking if a real function $g$ is identically zero is not decidable.*

*Proof.* To see this consider a function $g$ which is identically zero. Let the Turing machine $T$ query the points $x_1, x_2, \ldots,$. If we have another continuous function $g'(x) = g(x)$ at all points $x_1, x_2, \ldots,$ but $g'(x) \neq 0$ for some $x \notin \{x_1, x_2, \ldots, \}$. Then the Turing machine cannot output this function $g'$ as not being identically zero. $\qquad \square$

**Lemma 2.4.** *There is no algorithm to decide if a point $x_k$ is a $\epsilon-$ approximation to the global optima.*

*Proof.* Now consider the problem of deciding if a (non-convex) negative function $f$ is identically zero. This problem is unsolvable by Lemma 2.3. Since the function is negative, it is identically zero if and only if the global minimal value is zero. Similarly the function $f'(x) = \max\{0, f(x) + \epsilon\}$ is identically zero if and only if the $\epsilon$-approximation to the global minimal value is zero. Thus, we have a reduction from the problem of deciding if a function is identically zero to the problem of deciding if the global minimal value is zero. Thus our problem is unsolvable by Lemma 2.3. ☐

**Lemma 2.5.** *There is no algorithm to decide if a value $f(x_k)$ is a $\epsilon-$ approximation to the global optima.*

*Proof.* The proof is similar to the proof of 2.4. Consider the problem of finding the optimal point $x^*$ and its $\epsilon$-approximation. As we have access to $f(x)$ only from the function oracle, this problem is same as the one in Lemma 2.4. ☐

**Theorem 2.6.** *We assume the objective function we wish to minimize is known by its oracle. There is no algorithm which can compute the $\epsilon$-approximate optimal point (and optimal value) of a continuous, non-convex objective function on a compact domain.*

*Proof.* Let $x'_k$ be any point of some finite precision length $n_k$ such that $|f(x^*) - f(x'_k)| < \epsilon$. Such a point exists by Lemma 2.1 i.e., the set $G^f_{\epsilon,n_k}$ is non-empty for $\epsilon > 0$. Suppose that we have an algorithm to find a $\epsilon/2-$ approximation point $x'_k$. This length $n_k$ can increase with $k$.

Now for any point $x_k \in C$ we can say it is $\epsilon$ close to optimum if $|f(x_k) - f(x'_k)| < \epsilon/2$ else it is not. Thus we have an algorithm to decide if a point is an $\epsilon-$ approximation to the global optima or not. This is a contradiction to Lemma 2.4. Thus for a $\epsilon > 0$, there is no algorithm to find a $\epsilon-$ approximation to the global optima. The proof for optimal value similarly follows Lemma 2.5. ☐

**Corollary 2.7.** *The problem of checking whether local minima $z$ is global is also not decidable.*

*Remark* 2.8. Even in presence of higher order oracles, i.,e oracles which give derivatives of the function, the reduction from the problem of deciding if $f$ is identically zero remains. Hence global optima even in presence of these higher-order function oracle is not computable.

*Remark* 2.9. As we mentioned before, our result is for algorithms having access to the function oracle. This is different from the setting of computable function and reals studied in computable analysis. (Pour-El & Richards, 1989)

*Remark* 2.10. The finite-precision assumption is not restrictive. If there is an algorithm to find a general real number which is $\epsilon-$ approximation, we can take the first $k$ digits which gives $\epsilon-$ approximation to get a finite precision approximation.

*Remark* 2.11. The proof does not hold for finding local minima as the (negative) function need not be zero if its local optima is zero.

## 3 GLOBAL OPTIMA PROPERTY

In this section, we see a simple property a function satisfies if and only if the global optima is computable.

**Definition 3.1.** For a given function $f$, let $P^f$ be a first order predicate $P^f(x, y)$ defined as

$$P^f(x, y) = \begin{matrix} \text{True if } f(x) \le f(y), \\ \text{False o.w.} \end{matrix}$$

**Definition 3.2.** Let $Q$ be any first order (3-ary) predicate $Q(\zeta, x, y)$. We say that $P^f \subset Q$ if there exists a $\zeta \in \mathbb{R}$ such that,

$$P^f(x, y) = Q(\zeta, x, y), \text{ for all } x, y.$$

**Definition 3.3.** We say a property $Q(\zeta, x, y)$ is True if there exists a $\zeta$ such that it is True for all $x, y \in C$. And it is computable if for any $x$ and $y$, $\zeta$ in the definition of $Q$ can be computed.

We now prove the following:

**Theorem 3.4.** *The global optimal value (or optimizer) of a function $f$ can be approximated to any accuracy if there exists a computable predicate $Q$ which is True and $P^f \subset Q$. Further, if the global optimal value (or optimizer) is computable then there exists such a computable predicate which is True and $P^f \subset Q$.*

*Proof.* First note that the property $Q$ can be computed. That is we can compute a $\zeta$ such that $P^f \subset Q$. We know that there exists a $x$ such that for all $y$, $P^f(x, y)$ is True. If there exists a computable $Q$ with $P^f \subset Q$ then the tuple $(P^f(x, y), \zeta, x, y)$ is computable for any $x$ and $y$. From this and the fact that inverse of a surjective recursive function is recursive, we have: given a $y_0$ and a $\zeta$ we can compute a $x_0$ such that $P^f(x_0, y_0) = Q(\zeta, x_0, y_0)$ is True. Now take $y_1 = x_0$ and continue this process to obtain $x_1$. This sequence of $\{x_k\}$ converges to the the optima $x^*$ as $P^f(x, y)$ defines a partial order with $f(x_k) \leq f(x_{k-1})$. And from this sequence $\{x_k\}$ we can approximate the global optimizer to any accuracy.

For the second part, assume the global optimizer $x^*$ is computable. Now take $\zeta = ||x^*||$. This is computable as $x^*$ is. It is easy to see that the predicate $Q(||x^*||, x, y)$ is True, computable and $P^f \subset Q$. And since the function is known by its oracle, the optimal value can be found only from the optimizer. $\square$

*Remark* 3.5. Lipschitz continuity is an example of such a global property. Let $Q(L, x, y)$ be

$$|f(x) - f(y)| \leq L \parallel x - y \parallel, \text{ for all } x, y \in C.$$

And here the number $\zeta$ is the Lipschitz constant $L$. Let the set of functions on some compact domain $C$ satisfying the Lipschitz property be denoted by $\mathcal{L}$. It is known that if the Lipschitz constant or an upper bound to it is known then the global optima for this class of functions $\mathcal{L}$ can be approximated to any accuracy. (For example refer to Theorem 1.1.2 of (Nesterov)). Another example of a global property is bounded derivatives. If a bound on the gradient is known then the global minima can be computed to any accuracy.

## 4   AN ALGORITHM WITH KNOWN BASIN OF ATTRACTION

In this section, we also assume the function $f$ to be differentiable. Let us denote the gradient by $\nabla f(x)$. The algorithm takes as input the lower bound $m$ on the basin of attraction of the global minimizer. By basin of attraction we mean the following: if we let the initial point to be in the hypercube of length $m$ in all co-ordinates, i.e., in the basin of attraction around the global minima $B_m(x^*)$ then the gradient descent algorithm will converge to the minima. This is another example of a global optima property (independent of the last argument),

$$Q(m, x, \cdot) := \text{ If } x \in B_m(x^*) \text{ and } x \neq x^* \text{ then } \nabla f(x) \neq 0.$$

Note that there exists a $m$ such that $Q(m, x, \cdot)$ is True. If this basin of attraction size $m$ can be computed or its lower bound known then we can compute the global minima (Theorem 3.4) to any approximation. In this section, we given an algorithm which converges to the global minima if this basin length $m$ or its lower bound is known.

The algorithm finds the point $z_k$ where the function takes a minimum amongst all points at a distance of $m$ from each other and does a gradient descent step from the point $z_k$. The algorithm outputs an $\epsilon$-approximation to the global optima (see Theorem 5.4 for the value of $\epsilon$). In this algorithm for simplicity, we do not consider line searches and use constant step-size $t > 0$. The figure 1 shows the gradient descent step taken at the point which has the minimum function value amongst all the points in the grid.

We note the similarity of our algorithm with the one considered in paper (D'Helon et al., 2007), where the basin of attraction of global minimizer is first found by searching then a gradient descent is performed. In our algorithm, these two steps are interleaved. The major issue with their algorithm is that they assume the value of the global minima is known which they assume to be zero. But this need not be known in real-world problems. This assumption is not needed with our approach. Moreover, we have formally shown the convergence of our algorithm.

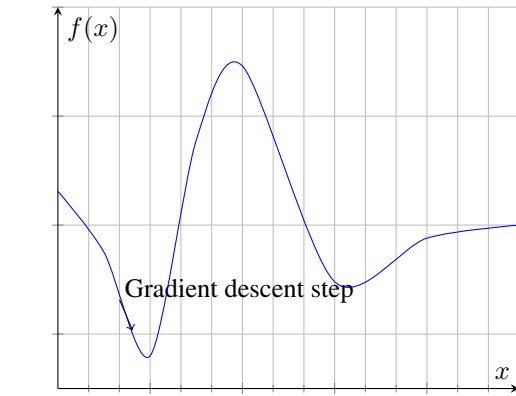

Figure 1: The function $f$ to be minimized. Gradient descent step is shown for the interval where the function value is minimum. This interval is a subset of the basin of attraction of global minima.

---

**Algorithm 1** Global Optimization Algorithm

---

**Input:** Function $f$ and a lower bound $m$ on length of a hypercube contained in basin of attraction of global minimizer of $f$

**Output:** An $\epsilon$-approximation to the global minima

1: Let $C = [a, b]^d$. For simplicity we let the interval $[a, b]$ to be the same in all dimensions.
2: Set $y_0 = [a_1^0, \ldots, a_d^0]$ where $a_i^0 = a$ for all $i = 1, \ldots, d$. And set $y_j = y_{j-1} + m$ for $j = 1, \ldots, (b-a)/m$. Let $x_0 = z_0 = \arg\min_{j=0,\ldots,(b-a)/m}\{f(y_j)\}$.
3: **while** $k = 1, \ldots, \mathcal{L}$ **do**
4:     Set $y_0 = [a_1^k, \ldots, a_d^k]$ where $a_i^k = a_i^{k-1} - t\nabla f(z_k)$ for all $i = 1, \ldots, d$. And set $y_j = y_{j-1} + m$ for $j = 1, \ldots, (b-a)/m$.
5:     As before let $z_k = \arg\min_{j=0,\ldots,(b-a)/m}\{f(y_j)\}$.
6:     Update $x_k = z_k - t\nabla f(z_k)$.
7:     $k = k + 1$
8: **end while**

---

## 5   Convergence Analysis

We show the convergence of the algorithm given in the preceding section. We make the following assumption.

**Assumption 5.1.** *The function $f$ is twice differentiable. The gradient of $f$ is Lipschitz continuous with constant $0 < L < 1$, i.e.,*

$$\| \nabla f(x) - \nabla f(z) \|_2 \leq L \| x - z \|_2 .$$

*That is we have $\nabla^2 f(x) \preceq LI$.*

We first state the following lemma used in the proof of the convergence theorem.

**Lemma 5.2.** *Assume that the function $f$ satisfies Assumption 5.1 and the step-size $t \leq 1/L$. We also assume that the global minima $x^*$ is unique. Then there exists a constant $R > 0$ such that for all balls $B(x^*, r)$ with radius $r < R$, there is a $M_r > 0$ and that the iterates of the algorithm $\{x_k\}$ remains in this ball $B(x^*, r)$ asymptotically, i.e., $x_k \in B(x^*, r)$ for $k \geq M_r$.*

*Proof.* From assumption 5.1 we have that $\nabla^2 f(x) - LI$ is negative semi-definite matrix. Using a quadratic expansion of $f$ around $f(x^*)$, we obtain the following inequality for $x \in B(x^*, r)$

$$f(x) \leq f(x^*) + \nabla f(x^*)^T (x - x^*) + \frac{1}{2}\nabla^2 f(x^*) \| x - x^* \|_2^2$$

$$f(x) \leq f(x^*) + \frac{1}{2}L \| x - x^* \|_2^2 \tag{1}$$

Since $x^*$ is a global minima we have $f(x^*) \leq f(x)$ for all $x \in C$. Let $\tilde{x}$ be any local minima which is not global minima. Hence $f(\tilde{x}) = f(x^*) + \delta_{\tilde{x}}$. Now let $\delta = \min_{\tilde{x}} \delta_{\tilde{x}}$. Since $\tilde{x}$ is local minima but not global minima we have $\delta > 0$. Take $R > 0$ such that for any $x \in B(x^*, R)$,

$$\frac{L}{2} \| x - x^* \|_2^2 \leq \frac{\delta}{2}$$

or that $R \leq \frac{\delta}{L}$. Now we have from equation equation 1

$$f(x) \leq f(x^*) + \frac{\delta}{2},$$

for $x \in B(x^*, R)$. That is we have shown there exists a $R > 0$ such that for any $x \in B(x^*, R)$,

$$f(x) \leq f(\tilde{x}). \tag{2}$$

Now we observe the following:

1. from equation equation 2 we can see that no other local minima can have a value $f(\tilde{x})$, lower than the function value in this ball $B(x^*, R)$

2. for sufficiently small step-size $t \leq 1/L$, the function value decreases with each gradient step (see equation equation 3 in proof of Theorem 5.4)

That is if $x \in B(x^*, R)$, the iterates in the algorithm can not move to another hypercube around some local minima $\tilde{x}$. Or that for all $r < R$ there exists $M_r > 0$ such that for $k \geq M_r$ the iterates remain in the ball $B(x^*, r)$ around $x^*$. $\qquad\square$

**Theorem 5.3.** *Let $x^*$ be the unique global minimizer of the function $f$. We have for the iterates $\{x_k\}$ generated by the algorithm*

$$\lim_{k \to \infty} f(x_k) = f(x^*).$$

*Proof.* Now from Lemma 5.2 we have $R > 0$ such that for all $r < R$ there exists $M_r > 0$ with $x_k \in B(x^*, r)$ for $k \geq M_r$. From the algorithm we also know that the function value decreases with each iteration. Thus we see that the sequence $\{f(x_k)\}$ converges as it is monotonic and bounded. Take a sufficiently small $r < R$, such that $B(x^*, r)$ lies in the basin of attraction. Hence we also have that $\lim_{k \to \infty} f(x_k) = f(x^*)$ as in the basin of attraction around the global minima the gradient descent converges to the minima. $\qquad\square$

**Theorem 5.4.** *Let $x^*$ be the unique global minimizer of the function $f$. For simplicity denote $M = M_r$. Let step-size $t \leq 1/L$ where $L$ is Lipschitz constant of the gradient function in Assumption 5.1. If we also assume that the function is convex in the ball $B(x^*, r)$ we can show that at iteration $k > M$, $f(x_k)$ satisfies*

$$f(x_k) - f(x^*) \leq \frac{\| x_M - x^* \|_2^2}{2t(M - k)}.$$

*That is the gradient descent algorithm converges with rate $O(1/k)$.*

*Proof.* Consider the gradient descent step $x_{k+1} = z_k - t\nabla f(z_k)$ in the algorithm. Since the iterates remain in a ball around a global minima asymptotically, we have from Lemma 5.2 for $k \geq M_r$, $z_k = x_k$. Now let $y = x - t\nabla f(x)$, we then get:

$$f(y) \leq f(x) + \nabla f(x)^T (y - x) + \frac{1}{2}\nabla^2 f(x) \| y - x \|_2^2$$

$$\leq f(x) + \nabla f(x)^T (y - x) + \frac{1}{2}L \| y - x \|_2^2$$

$$= f(x) + \nabla f(x)^T (x - t\nabla f(x) - x) + \frac{1}{2}L \| y - x \|_2^2$$

$$= f(x) - t \| \nabla f(x) \|_2^2 + \frac{1}{2}L \| y - x \|_2^2$$

$$= f(x) - \left(1 - \frac{1}{2}Lt\right)t \| \nabla f(x) \|_2^2 .$$

Using the fact that $t \leq 1/L$, $-\left(1 - \frac{1}{2}Lt\right) \leq -\frac{1}{2}$, hence we have

$$f(y) \leq f(x) - \frac{1}{2}t \parallel \triangledown f(x) \parallel_2^2 . \tag{3}$$

Next we bound $f(y)$ the objective function value at the next iteration in terms of $f(x^*)$. Note that by assumption $f$ is convex in the ball $B(x^*, r)$. Thus we have for $x \in B(x^*, r)$,

$$f(x) \leq f(x^*) + \triangledown f(x)^T (x - x^*)$$

Plugging this into equation equation 3 we get,

$$f(y) \leq f(x^*) + \triangledown f(x)^T (x - x^*) - \frac{t}{2} \parallel \triangledown f(x) \parallel_2^2$$

$$f(y) - f(x^*)$$

$$\leq \frac{1}{2t}\left( 2t \triangledown f(x)^T (x - x^*) - t^2 \parallel \triangledown f(x) \parallel_2^2 \right)$$

$$\leq \frac{1}{2t}\left( 2t \triangledown f(x)^T (x - x^*) - t^2 \parallel \triangledown f(x) \parallel_2^2 \right.$$

$$\left. - \parallel x - x^* \parallel_2^2 + \parallel x - x^* \parallel_2^2 \right)$$

$$\leq \frac{1}{2t}\left( \parallel x - x^* \parallel_2^2 - \parallel x - t\triangledown f(x) - x^* \parallel_2^2 \right)$$

By definition we have $y = x - t\triangledown f(x)$, plugging this into the previous equation we have

$$f(y) - f(x^*) \leq \frac{1}{2t}\left( \parallel x - x^* \parallel_2^2 - \parallel y - x^* \parallel_2^2 \right) \tag{4}$$

This holds for all gradient descent iterations $i \geq M$. Summing over all such iterations we get:

$$\sum_{i=M}^{k} \left( f(x_i) - f(x^*) \right)$$

$$\leq \sum_{i=M}^{k} \frac{1}{2t}\left( \parallel x_{i-1} - x^* \parallel_2^2 - \parallel x_i - x^* \parallel_2^2 \right)$$

$$= \frac{1}{2t}\left( \parallel x_M - x^* \parallel_2^2 - \parallel x_k - x^* \parallel_2^2 \right)$$

$$\leq \frac{1}{2t}\left( \parallel x_M - x^* \parallel_2^2 \right).$$

Finally using the fact that $f$ is decreasing in every iteration, we conclude that

$$f(x_k) - f(x^*) \leq \frac{1}{k} \sum_{i=M}^{k} \left( f(x_i) - f(x^*) \right)$$

$$\leq \frac{1}{2t(M-k)} \parallel x_M - x^* \parallel_2^2 .$$

$\square$

*Remark* 5.5. If the global minima $x^*$ is not unique, then the algorithm can oscillate around different minima. If we assume that the function is convex in a small interval around all these global minima, then we can show that the algorithm converges to one of the minimum points $x^*$. In addition like in the previous theorem we can also show that the convergence is linear.

*Remark* 5.6. We have not considered momentum based acceleration methods which fasten the rate of convergence in this paper.

Table 1: Various Benchmark Functions for Global Optimization

| Name | Global Minimum | Search Domain | Step-size | Lower bound on the basin |
|---|---|---|---|---|
| Rastrigin Function | $f(0, \ldots, 0) = 0$ | $-5.12 \leq x_i \leq 5.12$ | 0.0001 | 0.5 |
| Ackley Function | $f(0, 0) = 0$ | $-5 \leq x, y \leq 5$ | 0.0001 | 0.1 |
| Sphere Function | $f(0, \ldots, 0) = 0$ | $-\infty \leq x_i \leq \infty$ | 0.001 | 0.3 |
| Rosenbrock Function | $f(1, \ldots, 1) = 0$ | $-\infty \leq x_i \leq \infty$ | 0.001 | 0.5 |
| Beale Function | $f(3, 0.5) = 0$ | $-4.5 \leq x, y \leq 4.5$ | 0.0005 | 0.3 |
| Booth Function | $f(1, 3) = 0$ | $-10 \leq x, y \leq 10$ | 0.005 | 0.3 |

## 6 EXPERIMENTAL RESULTS

There are many standard test functions for testing global optimization algorithms. Some have many local optima, some are bowl/valley shaped etc. Let us consider one such function known as the Beale function. It is defined on a real plane as a rectangular region $B = [(-4.5, -4.5), (4.5, 4.5)]$, $f : B \to \mathbb{R}$ as in Table 1. The function has global optimum at $x^* = (3, 0.5)$ and $f(x^*) = 0$. We can also find the local optima of the function by finding the gradient and setting it to zero. One such local optima is at $(0, 1)$. Gradient descent algorithm can be used to find these local minima.

Recall our setting, we do not know the function $f(x)$ analytically. In other words the function is known through an function oracle. And our result says that starting at any arbitrary point in this region $B$, there is no algorithm which can approximate the global minima. The results holds even in the presence of higher-order function oracles unless bounds on derivatives are known.

We present some numerical results. We tested the algorithm on standard benchmark functions shown in Table 1. We show the plots of the function value as iteration proceeds for each of these functions. For Rastrigin, sphere and Rosenbrock functions the dimension was set to 20. We see from these plots that the algorithm converges to the optimum for each of these functions as expected. Table 1 also gives the step-sizes and lower bound on the basin of attraction set used for each of these functions.

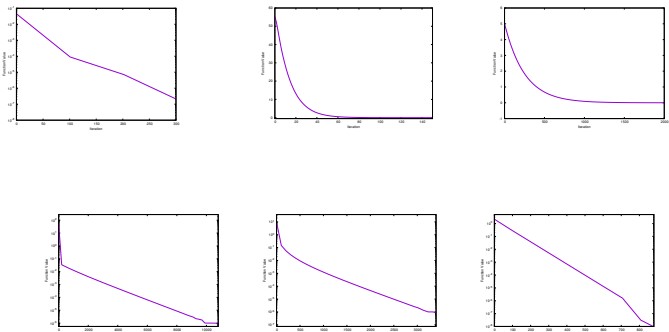

Figure 3: Convergence to Optimum for Ackley, Rastrigin, Rosenbrock, Booth, Beale and Sphere Functions (Clockwise from top left)

## 7 CONCLUSION

We have proved that there is no algorithm with only oracle access to the function to compute a $\epsilon-$ approximation to the optimizer (and optimal value) of a nonconvex function $f$. This result holds even if the function has higher-order derivatives. We have given a necessary and sufficient property a function has to satisfy if its global minima is computable. Then, we give basin of attraction as an example of this property and an algorithm which approximates the global minima with this property. Finally, some numerical results were presented.

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
