# OpenReview forum: "Approximating Optima of Nonconvex Functions"
_ICLR.cc/2025/Conference — ICLR 2025 Conference Withdrawn Submission_

### Official Review · Reviewer_fbP8 · 2024-10-27

**Soundness:** 3
**Presentation:** 1
**Contribution:** 1
**Rating:** 3
**Confidence:** 4

**Summary:**

The paper studies the computability of global minima of continuous nonconvex functions.

It is shown that global minima are not computable. Afterwards, a certain condition is suggested under which they are.

**Strengths:**

The results seem correct, and the proofs are pretty easy to follow.

**Weaknesses:**

This paper is very clearly unsuitable for acceptance in my opinion, for several reasons:

## Writing quality

The first thing that strikes the reader is that this paper is poorly written. For example, the opening paragraph already has several typos (".  .") and grammatical errors ("Global minima is..." - minima is plural, "by extreme value theorem" without "the" etc.) which appear everywhere in the paper.

Yet I do not mean this just in terms of English and typos. The literature of nonconvex optimization and computability aspects thereof is very rich, and nearly no references and comparisons are given. Even the few references that are given are treated very oddly. For example, in the second paragraph it is written that the considered setting "is different from... (and) "more general than..." - why? how so? what do the other papers even consider or study? This is never explained.

The section "Real-worlds applications" just goes over some of the most generic problems in which optimization is even applied, such as "supervised learning". It is not clear how this relates at all to the results in this paper.

And so on...

## Novelty?

Related to the fact that prior work is not really discussed, I would argue that the main results are nearly trivial, and are "folklore".

The main lower bound is proved by showing that a nonconvex function can hide function jumps in arbitrarily small neighborhoods, since no Lipschitz bound is assumed, which is trivial in my opinion. I strongly believe optimizaiton experts are well-aware of this, and multiple variations along these lines is stated throughout the literature.


To sum-up, this paper is clearly subpar for acceptance to ICLR in my opinion.

**Questions:**

-

---

### Official Review · Reviewer_wMLm · 2024-10-28

**Soundness:** 3
**Presentation:** 1
**Contribution:** 1
**Rating:** 1
**Confidence:** 3

**Summary:**

The paper presents non-computability results for non-convex functions in the context of Turing machines.

**Strengths:**

The paper deals with computability issues in optimizing non-convex functions. This problem can be relevant to many real-world applications of optimization and certainly of interest for the ICLR community.

**Weaknesses:**

Presentation: The presentation is problematic. There are various linguistic issues and whole phrases and explanations that are totally unreadable to me. The paper looks like it was finished in rush and is certainly far below the ICLR publication standards presentation-wise. The discussion of related literature is poor. With a quick look in the cited paper by Lee et al., I can see much related previous work, which is not discussed at all in the current work.

Examples of linguistic issues:

1. All sentences starting with "And".
2. "We note that we consider an oracle setting, where the function values are given by an oracle. This is
different from the computablity of optima computable real functions studied for example in (Pour-El
& Richards, 1989)".
3. "We show more in this paper,
that this set S is not computable. This is much stronger than saying they it is intractable."
4. "We show that Lipschitz constant is an example of
a property a function must satisfy if its approximate optima is computable."
5. "We give an example of global optima property-basin of attraction. And if this is known, we
give an algorithm which converges to the global optima."

These are just a small sample from the first page. The paper needs certainly a lot of work to become of publication quality.

Contribution: I do not actually grasp the novelty of the contribution of this work. It seems to me that the main result is trivial. More precisely,  Lemma 2.4 states that there is not algorithm to decide whether a point is an $\epsilon$-approximation of the global optimum of a non-convex function if no other information except an oracle that computes the function values is given. Why this is surprising? Lemma 2.5 is essentially the same result with Lemma 2.4. Then, the main result (Theorem 2.5) states that there is no algorithm to compute an approximation of the global optimum. This looks trivial to me as the most expensive algorithm one could sketch using only access to function values, is to create a huge grid which will result in some point sufficiently close to the global optimum and then evaluate the function at all these points. This is discussed in the first chapter of the cited textbook of Yurii Nesterov. However, without additional information to relate function values and distances of points to the optimum (like Lipschitz continuity with known Lipschitz constant as the authors mention, btw this is Theorem 1.1.1 and not 1.1.2), there is no way to show that the point with the smallest value is indeed close enough to the optimum. I don't see what is groundbreaking here.

**Questions:**

I would like the opinion of the authors in the following:

Why computability is an important notion to study in the first place, and not just restrict ourselves in the question of tractability? I understand why Yurii Nesterov discusses algorithms that work using huge grids in his textbook, but I'm not sure that the optimization community nowadays should necessarily care about problems that can be solved with such expensive algorithms. On the other hand, tractability is much closer to what optimization problems can be solved practically.

---

### Official Review · Reviewer_pBoh · 2024-11-02

**Soundness:** 2
**Presentation:** 1
**Contribution:** 2
**Rating:** 3
**Confidence:** 2

**Summary:**

This paper seeks to find properties of function so that their optima and optimizer are not computable in the sense they described.

**Strengths:**

I do not know the strength of the paper; it needs to be more precise and cared enough before being reviewed carefully.

**Weaknesses:**

There are lots of typos:
- line 022 (two dots to end the sentence).
- "This is much stronger than saying they it is intractable."
- "Then we give a simple algorithm which converges to the global optima when this is known" -> what is "this"?
- "We give an example of global optima property-basin of attraction. And if this is known," -> what is "this"?
- etc...

The authors should not give the reviewers the impression that the paper was written in haste; otherwise, we will not take time to read and assess it.

Advice: Put as much context as possible. For instance:
- "We also see that if the Lipschitz constant is not known, the (approximate) optima is not computable." The "see" is not an argument; the authors should refer to proof or a paper proving that.
- Different from "computable real-functions setting" -> The fact that it is different is irrelevant; we would like to know why this new setting would be more relevant or better quantify a set of functions encountered in practice, etc...
- "We now start with the definition of the standard Turing machine here." -> Add some context: Why does it matter, how is this indeed structural enough to define the notion of computable, why did the authors choose this notion of computable, and how does this notion translate to something useful for the ML/AI community?
- etc..

**Questions:**

N.A.

---

### Official Review · Reviewer_hsSY · 2024-11-03

**Soundness:** 1
**Presentation:** 1
**Contribution:** 1
**Rating:** 3
**Confidence:** 5

**Summary:**

The paper attempts to study lower bounds for nonconvex optimization. It claims that its contributions are to give negative answers to the following questions about any continuous nonconvex function $f$.

(1) Can a Turing machine with zero-order function access to $f$ compute its global optimum?

(2) Can a Turing machine with ZO access to $f$ compute $x^\star$ (the global minimizer of $f$)?

(3) Can a Turing machine with ZO access to $f$ compute an $\varepsilon$-approximation to $f(x^\star)$?

The paper claims that these contributions are new.

**Strengths:**

The paper attempts to study lower bounds for nonconvex optimization. It claims that its contributions are to give negative answers to the following questions about any continuous nonconvex function $f$.

(1) Can a Turing machine with zero-order function access to $f$ compute its global optimum?

(2) Can a Turing machine with ZO access to $f$ compute $x^\star$ (the global minimizer of $f$)?

(3) Can a Turing machine with ZO access to $f$ compute an $\varepsilon$-approximation to $f(x^\star)$?

The paper claims that these contributions are new.

**Weaknesses:**

From my understanding of the paper, there seems to be a lack of soundness in its claims, as I elaborate below.

----
The authors aim to establish that approximating the global minimum of a nonconvex function is non-computable, building this conclusion through Lemmas 2.3, 2.4, and 2.5, which feed into Theorem 2.6.

However, the problem to me appears to be in **Problem 1.8**, which asks whether a Turing machine with oracle access to a continuous, nonconvex function $f$ can compute the exact global minimum. This question is already known to be infeasible (see Nemirovskii and Yudin, 1983). Thus, the answer to **Problem 1.8** is immediately “no,” without requiring additional undecidability arguments.

The authors attempt to support their claim through a sequence of undecidability reductions, but it seems to me that those also have a logical flaw:

- **Lemma 2.3** asserts that deciding if a function is identically zero is undecidable. This is fine, because such a decision would require querying *every* possible point in the function’s domain.

- **Lemma 2.4** leverages Lemma 2.3 for a *special* case: it shows that if a positive function $f$ has a global minimum of zero, then determining whether an $\varepsilon$-approximation to this minimum is possible is equivalent to determining if $f$ itself is everywhere zero --- an undecidable problem by Lemma 2.3. This logic holds *within the specific setup*.

The authors then state **Theorem 2.6**, claiming that no algorithm can compute an $\varepsilon$-approximation of the global minimum for any continuous nonconvex function given oracle access to $f$. Here, they make a **critical leap in logic**: they extrapolate, without justification (and, in fact, incorrectly) the undecidability result from Lemma 2.4’s special case (a positive function with a minimum of zero) to all nonconvex functions.

This generalization is the misstep—there’s no basis for assuming that all nonconvex functions would exhibit the same undecidability. Many nonconvex functions do allow for $\varepsilon$-approximations of global minima.

Importantly, these reductions seem, to me, unnecessary. Since **Problem 1.8** is already infeasible, the undecidability arguments are redundant (and, further, seem to have logical flaws).

----

The authors also miss all the literature on lower bounds around nonconvex problems. I strongly recommend starting with the paper of Carmon, Duchi, Hinder, and Sidford (which by no means is the start of the study of this question, but is, in my view, a nicely written paper with good pointers to other literature and provided some major recent results and techniques).

**Questions:**

Please could the authors address the points in the Weaknesses section?

---

### Note · Authors · 2024-11-12

I have read and agree with the venue's withdrawal policy on behalf of myself and my co-authors.